# Drug classification with a spectral barcode obtained with a smartphone Raman spectrometer

Un Jeong Kim [1,10], Suyeon Lee[1,10], Hyochul Kim[1], Yeongeun Roh[1], Seungju Han [2], Hojung Kim[1], Yeonsang Park [3,4], Seokin Kim[5], Myung Jin Chung[6,7,8,9], Hyungbin Son[5] & Hyuck Choo [1] ✉

Measuring, recording and analyzing spectral information of materials as its unique finger print using a ubiquitous smartphone has been desired by scientists and consumers. We demonstrated it as drug classification by chemical components with smartphone Raman spectrometer. The Raman spectrometer is based on the CMOS image sensor of the smartphone with a periodic array of band pass filters, capturing 2D Raman spectral intensity map, newly defined as spectral barcode in this work. Here we show 11 major components of drugs are classified with high accuracy, 99.0%, with the aid of convolutional neural network (CNN). The beneficial of spectral barcodes is that even brand name of drug is distinguishable and major component of unknown drugs can be identified. Combining spectral barcode with information obtained by red, green and blue (RGB) imaging system or applying image recognition techniques, this inherent property based labeling system will facilitate fundamental research and business opportunities.

Miniaturization of optical spectrometers has been an active area of research because the demand for portable scientific and industrial characterization tools remains high[1–5]. Furthermore, smartphones are ubiquitous devices that provide numerous applications and services. Recently, many efforts have focused on converting smartphone cameras into optical spectrometers for mobile food inspection[6,7] beauty care[8], health care[9], and other applications[10–14]. In these cases, the image sensor of the smartphone detects optical signals from the object of interest−such as reflectance, fluorescence, and Raman emissions. Then, the smartphone's application processor (AP) and communication chip can together perform on-device or cloud-linked analysis[12], providing identification of specimens or evaluation of physical or chemical conditions.

Most research on smartphone-based spectrometers uses gratings as a dispersion component, assembled in an external optics module[6–13]. Gratings is an excellent optical component in spectrometer to disperse optical signals with high spectral resolution, but is not easy to minimize its form factor to fit into smartphone. To overcome this issue, mini spectrometers by replacing conventional grating with such as photonic crystals[14,15], metasurfaces[16–18], quantum dots[19] and silicone nanowires[20] integrated on charge coupled detector (CCD) or CMOS image sensors have been investigated. To calculate the input spectrum, $s(\lambda)$ out of measured intensity, $I(\mathbf{x})$ at the detector, numerical analysis needs to be done as expressed by the equation below due to its low Q-factor or complicated form of response function, $r(\lambda,\mathbf{x})$ at each pixel where $\mathbf{x}$ is the position of each

[1]Metaphotonics TU, Samsung Advanced Institute of Technology, Suwon, Gyeonggi-do 16419, Republic of Korea. [2]Machine Learning TU, Samsung Advanced Institute of Technology, Suwon, Gyeonggi-do 16419, Republic of Korea. [3]Department of Physics, Chungnam National University, Daejeon 34134, Korea. [4]Institute of Quantum Systems, Daejeon 34134, Korea. [5]School of Integrative Engineering, Chung-Ang University, Seoul 06974, Republic of Korea. [6]Department of Digital Health, Samsung Advanced Institute of Health Science, Sungkyunkwan University, Seoul 06355, Korea. [7]Department of Radiology, Samsung Medical Center, Sungkyunkwan University, Seoul 06355, Korea. [8]Department of Data Convergence and Future Medicine, Sungkyunkwan University School of Medicine, Suwon, Gyeonggi-do 16419, Korea. [9]Medical AI Research Center, Research Institute for Future Medicine, Samsung Medical Center, Seoul 06351, Korea. [10]These authors contributed equally: Un Jeong Kim, Suyeon Lee. ✉e-mail: hyuck.choo@samsung.com

pixel at the detector.

$$I(\mathbf{x}) = \int r(\lambda, \mathbf{x}) \cdot s(\lambda) d\lambda \qquad (1)$$

Thus, experimental results in the literature[14–20] have substantial limitations—especially in terms of capturing weak and high spectral resolution required for Raman signatures.

Due to the increasing online pharmacies and supply chain, counterfeit drugs have become threatening even to public health safety. This issue becomes more critical since increasing the online pharmacies and supply chain can provide blind spots for counterfeit or substandard drugs to be distributed into the public health market[21]. Current smartphone applications (such as DrugID, ID My Pill, Pill Identifier, Pill Finder, and Drug Info) can distinguish drug types and models either by entering the name, shape, color, and/or etched marks of the drugs; or by comparing the drug pill's RGB images (acquired with the camera) with the U.S. Food and Drug Administration database. The identification accuracy is insufficient due to similar appearance, absence in the database, or other technical issues. In this sense, Raman spectrum can provide valuable information on drugs, and there have been some researches in the literature on classifying drugs by Raman spectroscopy with the aid of machine learning[22–26]. Classifying pharmaceutical ingredients, and detection of newly emerging psychoactive substance and illicit drugs were demonstrated by partial least squares-discriminate analysis (PLS-DA)[22], principal component analysis (PCA)[23] and CNN[24], respectively. Detection of illicit drugs[25] or psychoactive drugs[26] were demonstrated even in human urine and finger marks to prevent patients from overdose or misuse of it by support vector machines (SVM) and PLS-DA, respectively.

We demonstrated smartphone based Raman spectrometer which are enough for drug classification. The Raman spectrometer is composed of 2D periodic array of band pass filters on the image sensor of a Samsung Galaxy Note 9, with a compact external Raman module.

Raman intensity map captured by the image sensor is defined as Raman spectral barcode by the analogy of conventional barcodes, machine-readable optical labels that enable location, identification, and/or tracking. As a demonstration, we experimentally investigated 54 commonly used drugs for diabetes, hyperlipidemia, hypertension, painkillers, and nutritional supplements; which frequently come in almost identical shapes, sizes, and colors. Since each spectral barcode of drug contains unique Raman signatures of the material, we conducted the identification of spectral barcodes of drugs with a convolutional neural network (CNN) embedded in the smartphone. In addition, identification accuracy can be further enhanced by information fusion with spectral barcode and conventional RGB images taken by the smartphone camera. Another advantage of spectral barcode-based classification is that we can identify chemical component of unknown drugs once other drugs with the same chemical component are in the database.

Integrating with AI capability in the smartphone spectrometer allows users to analyze the spectrum at various places and situations. This will enhance its portability and usability of smartphone spectrometer in numerous disciplines including drug classification. Our proposed concept of a CNN powered spectral barcode will facilitate many research and business opportunities for smartphone spectrometers.

## Results

### Smartphone Raman spectrometer and spectral barcode

Figure 1 shows schematics of the smartphone Raman spectrometer and spectral barcode; which is the 2D Raman intensity map acquired with the smartphone Raman spectrometer, and an artificial intelligence algorithm embedded in the smart phone for classification. Raman signals are generated and collected by a compact external module integrated with a 785 nm laser diode. The miniaturized external Raman module is attached to the rear-wide camera of the Samsung Galaxy Note 9, and its detailed optical components and configurations is

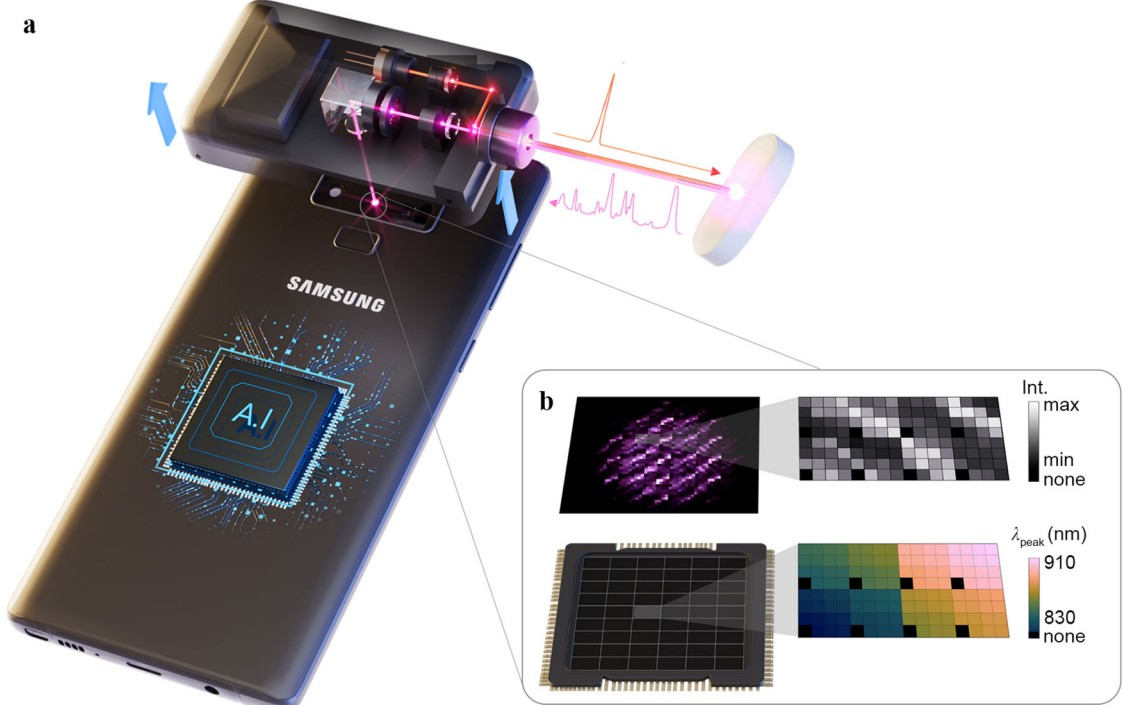

**Fig. 1 | Schematics of smartphone Raman spectrometer and data processing for analysis. a** The smartphone Raman spectrometer consisted of band pass filter arrays attached to a rear camera image sensor and an external attachable Raman module with a 785 nm laser diode. We embedded an artificial intelligence algorithm in the smartphone for classification. **b** Each band pass filter transmitted a specific wavelength and the captured image (containing spectral information) encoded as the spectral barcode. Color bar represents the peak wavelength of band pass filter ($\lambda_{peak}$).

shown in Supplementary Fig. 1 with a photograph. The Raman emission, which is excited by positioning the specimen at the focal point, i.e. contacting at the objective lens, simultaneously illuminates several sets of 128 channels (CHs) located near the center of the image sensor. For 120 CHs out of 128 CHs, its band pass filters transmit 120 distinct wavelengths in the range of 830–910 nm. The rest CHs are blocked by metal as position indicators exhibited as black squares in Fig. 1. The spectral width and transmission rate of the band pass filters range from 1–1.2 nm and 0.45–0.6, respectively (Supplementary Fig. 2). Each band pass filter consisted of a pair of $Si/SiO_2$ distributed Bragg reflectors (DBRs), its resonant wavelength is adjusted by the thickness of the Si cavity layer in the center[27,28]. The details of the filter structure and fabrication can be found in Methods. In Supplementary Table 1, the smartphone Raman spectrometer of this work is compared with miniaturized spectrometers which are controllable by android smartphones, or embedded in the smartphone[12,29,30]. The compared details are shown in the caption of Supplementary Table 1. As the role of the external module in this work is just to excite and collect Raman signals from the specimen without additional connecting electronic board to the smartphone, the smartphone Raman spectrometer becomes more compact and versatile with minimized external module.

From the image, a unique spectral barcode of the specimen is generated, which contains the Raman information of the sample. The Methods explains the detailed process to convert a raw image−acquired with the smartphone spectrometer−to a spectral barcode, a unique spectral identifier. Analogous to conventional barcodes, our work introduces a new concept of symbology to map spectral information into a spectral barcode: a set of multiple wavelengths, physical positions, and continuously variable transmitted Raman intensities at given wavelengths after normalization. Our spectral barcodes can express 1200 bits of information since 120 CHs deliver different wavelength information and one pixel of the image sensor encodes 10 bits. This is comparable with conventional 2D barcodes, which contain ca. 4000 bits of information depending on the symbology. The capacity of the encoding information of the spectral barcode can be enhanced by increasing the number of CHs or adapting sensor with higher dynamic range. Supplementary Fig. 3 shows examples of spectral barcodes of three drug pills that are similar in appearance: Glu-M SR for diabetes, Vitamin C 1000 mg Yuhan, and Tylenol 8 h ER Tab. Whereas they have a virtually indistinguishable appearance, one can easily distinguish their Raman spectra−obtained with a commercial spectrometer as well as corresponding Raman spectral barcodes obtained with our smartphone spectrometer. When comparing the Raman spectra obtained with the two measurements, blue squares indicate the Raman peaks or major spectral components of each drug and the corresponding locations in the Raman spectral barcode. Although the spectrum obtained with the smartphone Raman spectrometer exhibited a lower spectral resolution, it matched well with that of the commercial Raman spectrometer. The spectrum from the smartphone exhibited a slight shift in the peak locations (<1 nm), different relative peak intensities, and inter-peak spacing ($\Delta\lambda$) resulting from the spectral resolution. The full width at half maximum (FWHM) of the peaks corresponding to the C−O−C stretching bond at 861 nm and the C = C ring stretching bond at 903 nm of Vitamin C were 180% and 140% wider than those obtained with the commercial Raman spectrometer. Nevertheless, the narrowly spaced Raman bands at 899 nm (aryl CC stretch) and 903 nm (C = O stretch) were still well-resolved. The FWHM at these corresponding bands of 120 band pass filters ranged between 1 and 1.2 nm, and $\Delta\lambda$ was <1 nm.

## Drug classification using spectral barcodes

We demonstrated drug classification with a smartphone Raman spectrometer because this tool can provide important information in healthcare; for example, when distinguishing counterfeit from legal drugs, or choosing the correct drug pill among similar looking drug pills to prevent misuse. To overcome the issues of previous works as explained in the introduction, Raman spectroscopy provides molecular fingerprints and is suitable for identifying drugs by their chemical compositions and functions. We chose the most widely prescribed drugs for three common diseases (hypertension, diabetes, and hyperlipidemia) and three over-the-counter medicines (vitamin B6, vitamin C, and acetaminophen) for drug classification. Medical professionals prescribe amlodipine, losartan, and valsartan for hypertension; glimepiride and metformin for diabetes; and atorvastatin, rosuvastatin, and simvastatin for hyperlipidemia. Supplementary Fig. 4 shows the chemical structures of the major components, and Supplementary Fig. 5 shows the list of 58 drugs as well as their major components and RGB images. Supplementary Fig. 6 shows reference Raman spectra measured with a commercial Raman spectrometer at 785 nm excitation. Raman spectra of the same component exhibited the same Raman peaks, whereas the intensity of the background was quite different; or even new Raman peaks were evident at 810, 825, and 830 nm due to the additives in the drugs (such as atorvastatin and simvastatin for hyperlipidemia). Figure 2 shows representative spectral barcodes of 11 major components found in hypertension, diabetes, hyperlipidemia, and the other over-the-counter drugs. Spectral barcodes result from sharp Raman bands and broad fluorescence, which produce different patterns. Most of the spectral barcodes are readily distinguishable; but in some cases, drugs with different major components (for example, amlodipine, losartan, and simvastatin) need a classification algorithm to distinguish.

Figure 3 shows the schematics of data processing for drug classification based on spectral barcodes. When combined with CNN, Raman spectroscopy becomes a powerful tool for predicting the major components of drugs and even their brand identities. We used 54 drugs (1–54 in Supplementary Fig. 5) to train and test the neural network, and four drugs (A1–A4 in Supplementary Fig. 5) to prove the hypothesis that a CNN based on Raman spectral barcodes can properly recognize chemical components of drugs that are not in the database to train CNN. The details to obtain Raman spectral images are explained in the Methods. The statistical analysis of 42 Raman spectral images of Vitamin C to test CNN has been done as shown in Supplementary Fig. 7. The average value with standard deviation of normalized Raman intensity at each wavelength of the spectral barcodes is plotted. We used RGB images as additional information to improve drug classification accuracy by their brand name. The entire process, from the measurement (Raman spectral barcodes and RGB images) to the display of the results (types or brand names of the drugs), can be completed with a single device by using the pre-embedded CNN algorithm in the smartphone's AP. Among various classification algorithms such as Bayesian network, support vector machine (SVM), etc., we select CNN with a simplified Residual neural network (ResNet) architecture to identify the major component of drugs (based on a common CNN structure, including e.g. Alex neural network (AlexNet) and visual geometry group neural network (VGGNet), and implemented a shortcut(add) skipping convolution)[31]. This CNN is made up of one conventional residual block of ResNet, consisting of a convolution layer with batch normalization, add, and rectified linear unit (ReLu); and two fully connected layers produced after flattening (one with batch normalization and ReLu, and the other with batch normalization and softmax). ReLu is a common activation function in deep learning algorithms and returns a max (0, input), which provides a threshold in various parameters generated during the execution of the algorithm. The Methods describes details of the CNN architecture, training method, and database. To identify the brand name of each drug, we applied another CNN−simplified ResNet−followed by classification of the major component. The architecture of the CNN for identifying the brand name was similar to that of the CNN for classifying the major component, except the size of the fully connected

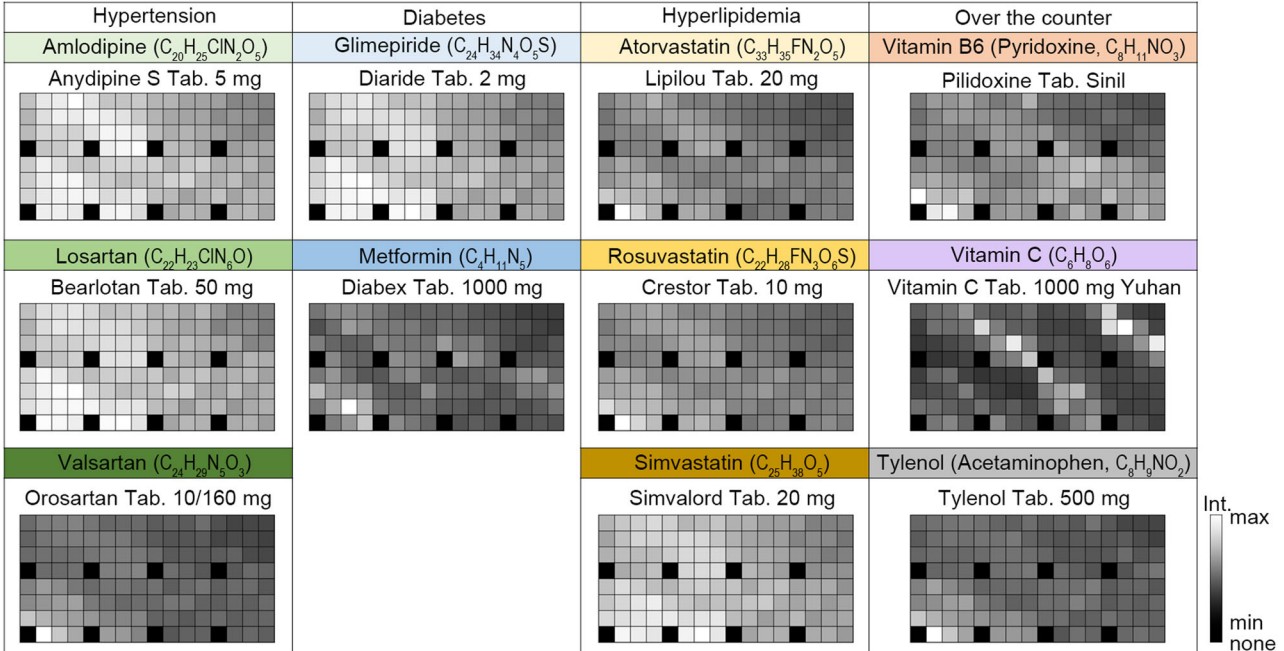

**Fig. 2 | Representative spectral barcodes of 11 major components of drugs.** Representative spectral barcodes of amlodipine, losartan, and valsartan for hypertension; glimepride and metformin for diabetes, atorvastatin, rosuvastatin, and simvastatin for hyperlipidemia; and vitamin B6, vitamin C, and Tylenol for over-the-counter drugs. Each panel also shows the specific brand names that correspond to the spectral barcodes for each major component.

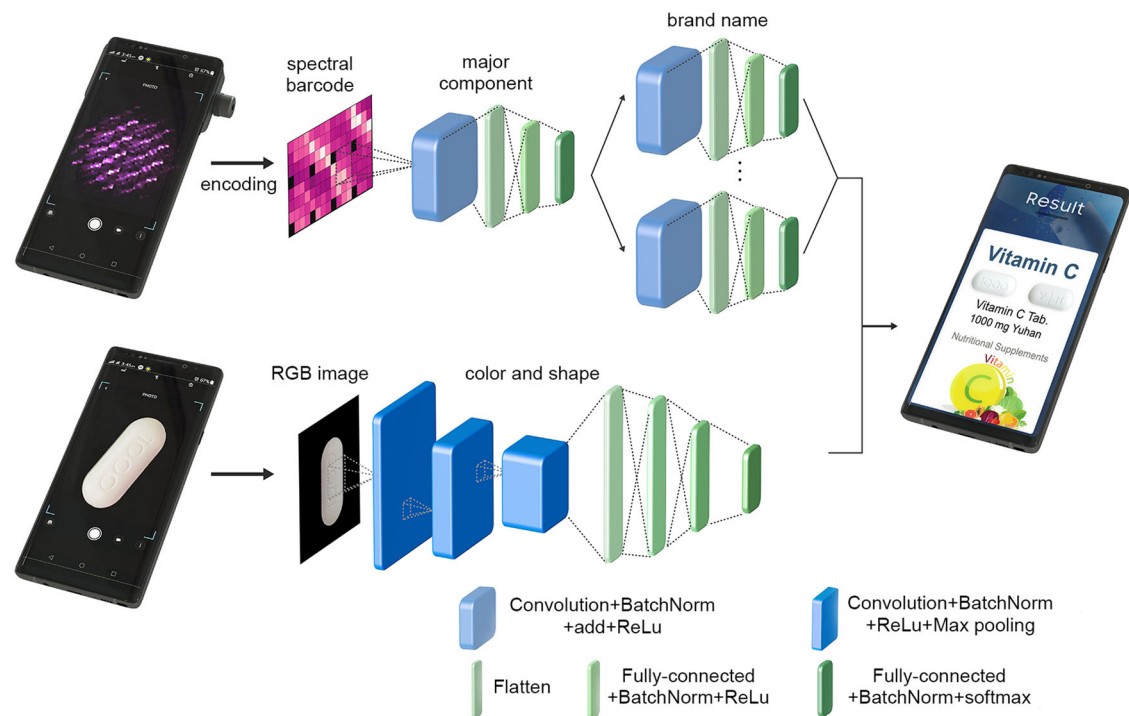

**Fig. 3 | Schematics of encoding spectral barcode and data processing to analysis.** Encoding the spectral barcode from the 2D Raman image of a drug as well as classification by major component or brand name with convolutional neural network (CNN). One conventional residual block of residual neural network (ResNet) (convolution layer with batch normalization, add, and rectified linear unit (ReLu)). Two fully connected layers followed by flattening (one with batch normalization and ReLu, and one with batch normalization and softmax as an activation function). We combined a CNN for the red, green and blue (RGB) images of the drugs taken by the smartphone camera as a tool to enhance the accuracy of drug classification. Schematics of CNN architecture (visual geometry group neural network (VGGNet)) for classifying the shape and color from the RGB image of the drug. The smartphone shows the results as an auxiliary classification tool.

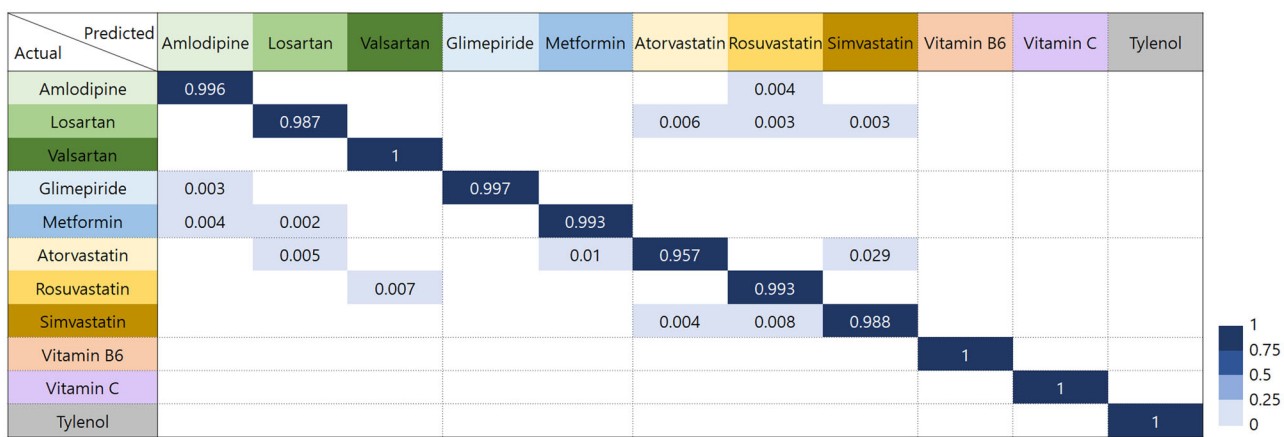

| Actual \ Predicted | Amlodipine | Losartan | Valsartan | Glimepiride | Metformin | Atorvastatin | Rosuvastatin | Simvastatin | Vitamin B6 | Vitamin C | Tylenol |
|---|---|---|---|---|---|---|---|---|---|---|---|
| Amlodipine | 0.996 | | | | | | 0.004 | | | | |
| Losartan | | 0.987 | | | | 0.006 | 0.003 | 0.003 | | | |
| Valsartan | | | 1 | | | | | | | | |
| Glimepiride | 0.003 | | | 0.997 | | | | | | | |
| Metformin | 0.004 | 0.002 | | | 0.993 | | | | | | |
| Atorvastatin | | | 0.005 | | 0.01 | 0.957 | | 0.029 | | | |
| Rosuvastatin | | | 0.007 | | | | 0.993 | | | | |
| Simvastatin | | | | | | 0.004 | 0.008 | 0.988 | | | |
| Vitamin B6 | | | | | | | | | 1 | | |
| Vitamin C | | | | | | | | | | 1 | |
| Tylenol | | | | | | | | | | | 1 |

Color scale: 1 / 0.75 / 0.5 / 0.25 / 0

**Fig. 4 | Confusion matrix of classifying the major component of 54 drugs.** Diagonal and off diagonal terms represent correct and wrong classification of drugs. Color scale bar for relative number at each cases out of total trials is shown in the right corner.

layers (since the size is related to the dimensions of the final result; the brand name of the drug).

**Classification results for the major component and brand name**
Figure 4 shows the confusion matrix for classifying the major chemical components of the drugs. The confusion matrix is for evaluating the performance in classification problems, comparing the actual class, and predicting the class with a classification algorithm. Diagonal and off-diagonal terms represent the correct and incorrect cases, respectively. Valsartan, vitamin B6, vitamin C, and Tylenol produced 100% accurate classification. The overall accuracy for 54 drugs major component was 99.0%. Additionally, we confirmed the expandability and effectiveness of the CNN for spectral barcodes by identifying four drugs as listed A1, A2, A3, A4 in Supplementary Fig. 5, (Glimel 3 mg, Dymit XR, Glucophase 1000 mg, and Metofol 500 mg). Even though these drugs are excluded in CNN training procedure (both training and validation set), the trained CNN accurately predicted the major components from the spectral barcodes once the spectral barcode of the same major components were in the database. Regarding Dymit, Glucophase, and Metofol, other 11 drugs with the same major components (metformin) were in the database. Regarding Glimel, eight drugs with glimepiride were in the database. The prediction accuracy for the major component of three drugs from metformin and one drug from glimepiride was obtained from 424 and 141 trials, respectively. Only one failure from metformin was confirmed, which corresponds to 99.8% accuracy for the major component prediction of unknown four drugs.

It is valuable to mention that broader spectral range could enhance the accuracy to predict its brand names by designing the band pass filter arrays on the image sensor to capture additional Raman features, for example, at 810, 825 and 830 nm associated with additives in the drugs as shown in Supplementary Fig. 6.

Classifications on various applications by CNN have been done using full spectrum of objects under interest obtained by benchtop or portable spectrometers using high signal-to-noise ratio (SNR) CCD and conventional grating[22,26] with high spectral resolution. The smartphone based Raman spectrometer using 120 filter arrays on CMOS image sensor produces lower spectral resolution but still high Q factor (>1 nm by FWHM). CMOS image sensors are highly efficient in power consumptions compared to CCD. Thus, the developed spectrometer on CMOS image sensor exhibits SNR and Q factor enough to classify drugs by Raman spectral barcode, and is suitable for lower power consumption.

It might occasionally be necessary to identify the names as well as brands of drugs that are in the same drug group because brand-specific additives or coatings can affect the behavior in the body, such as speed of absorption or allergic reaction. Figure 5 shows the spectral barcodes of three metformin drugs (Diabex 1000 mg, Dybis, and Glu-M SR) and their spectra. The squares of the same color indicate the Raman peaks which are from the same major chemical component, metformin. Higher fluorescence appears for Glu-M SR than Diabex as the overall intensity was high in the spectral barcodes. The accuracy in terms of classifying brand names remained still large: 79.5% (Supplementary Fig. 8 shows the confusion matrix for brand name from the spectral barcode.), since the additives or coatings provided increased fluorescence levels or additional Raman bands, which provide the distinguishability among drugs with the same major component. The accuracy of the CNN for differentiating one major component from the others was high, and thus misclassifying cases were most common among drugs with the same major components. (Supplementary Fig. 8).

**Information fusion with RGB image for classification accuracy enhancement**
The appearance of the drugs such as color and shape provides additional information for identification as RGB images taken by the smartphone camera exhibit various shapes and colors (Supplementary Fig. 5). We applied CNN with a commonly used VGGNet architecture[32,33] in conventional RGB imaging of drugs to recognize the shape (snowman, circle, ellipse, and pentagon/octagon) and color (blue, yellow, green, white, and pink) for higher recognition accuracy as well as brand name classification (Fig. 3). We achieved classification by subsampling (i.e. reduced data size) with a convolution layer, fully connected layer, and max pooling; and used ReLu as an activation function. Supplementary Fig. 9 and the Methods show the confusion matrix and the architecture of the CNN algorithm for the RGB images for classifying the shape and well as color. By additionally applying the CNN of RGB images as an auxiliary classification tool, the accuracy of identifying the exact brand name was slightly increased up to 83.2% (Supplementary Fig. 10 shows the confusion matrix). We designed the final CNN (for predicting the brand name) to use the product of the outputs from both CNNs as a combined method, treating them with equal importance. One could further optimize the prediction accuracy by adjusting the output ratio between two types of CNNs. One could also use the imprinted marks on the drugs in conjunction with proper image processing, and/or further subdivisions of shape and color for appearance recognition with the RGB images.

**Discussion**
In this work, we introduced the concept of the spectral barcode, obtained with a smartphone Raman spectrometer. Even with relatively lower spectral resolution and SNR due to the inherent properties of

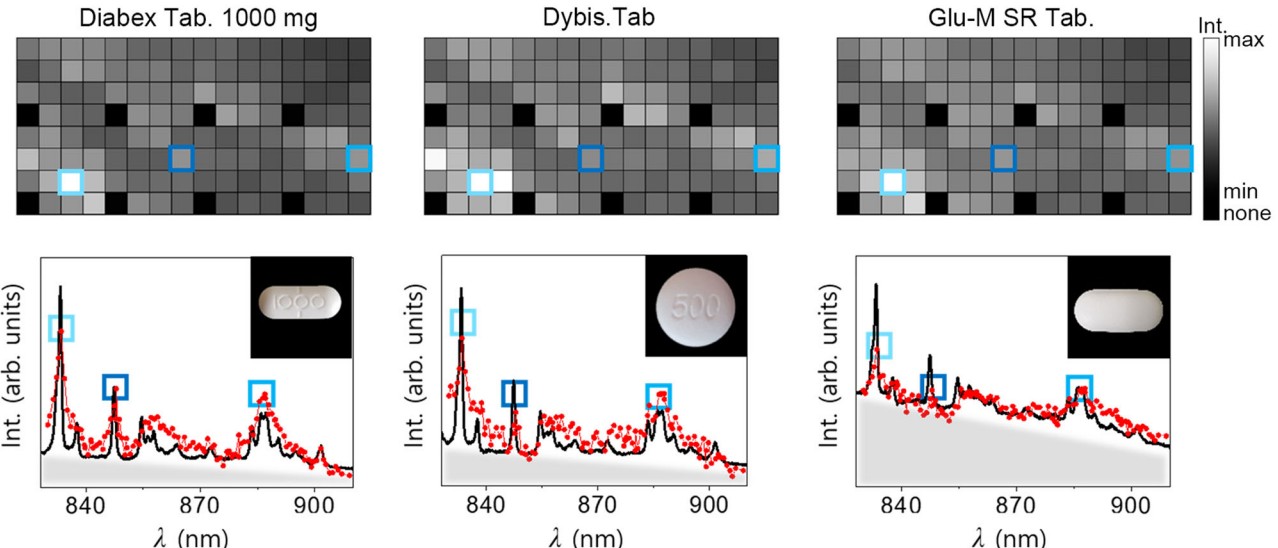

**Fig. 5 | Comparison of the spectral barcodes with same major component.** Three spectral barcodes from the same major component of drug category (metformin). Diabex 1000 mg, Dybis, and Glu-M SR shown from left to right. The Raman spectrum extracted from each barcode are below the spectral barcode, along with the reference Raman spectrum from a commercial spectrometer, indicated by a black solid line and red symbols connected by lines, respectively. The insets show the RGB images of each drug. The gray-shaded tapered areas indicate fluorescence or background, induced by additives or binders.

band pass filter arrays and CMOS image sensor compared with commercially available spectrometers installed with grating and CCD, the smartphone Raman spectrometer exhibits still high enough Q factor as portable spectrometer with high efficiency in terms of power consumption. Only external excitation and collection optics are needed to excite and collect Raman signals from the specimen without additional connecting electronic board to the smartphone. This makes the smartphone spectrometer more compact with minimized external module and versatile. Integrating with AI capability in the smartphone spectrometer makes the developed spectrometer more powerful. We demonstrated drug classification by spectral barcodes containing weak Raman signals with 99.0% and 79.5% accuracy for major component and brand name, respectively. By combining a CNN for RGB images of drugs, we increased the accuracy of brand name up to 83.2%.

There might be methods to increase the prediction accuracy to 100% by using additional information on drugs, such as etched marks or additional spectral features, and even upgrading the specification of the smartphone spectrometers such as spectral range, Q-factor, SNR and etc. In the measurement aspect for prediction accuracy enhancement, detection of major components under thick coating may be possible, for example, by introducing spatially offset Raman spectroscopy (SORS). Fluorescence could be separated from Raman signals of drugs by shifted-excitation Raman difference spectroscopy (SERDS). Moreover, systematic understanding how drug companies mix major components in collaboration with medical society is necessary to develop more powerful drug classification CNN.

In the future, by reducing the size of channel to one-pixel level and increasing the density of CH arrays, simultaneous measurement of spectral and morphological information of the object under interest can be achieved, which is called hyperspectral imaging, by using smartphone camera. This will extensively increase the portability and usability opening up new field in smartphone business.

## Methods
### Samples
Fifty four drugs for hypertension (Amlodipine, losartan, and varsartan), diabetes (metformin and/or glimepride), and hyperlipidemia (atorvastatin, rosuvastatin, and simavastatin) are provided by Samsung medical center and returned back for discarding after finishing the project. Vitamin B6 (plidoxine Tab. Sinil), Vitamin C (Vitamin C Tab. 1000 mg Yuhan),Tylenol Tab. 500 mg and Tylenol 8 h ER Tab. were purchased from local pharmacies in South Korea. The name, appearance and pharmaceutical companies of each drug can be found in Supplementary Fig. 5 as annotation.

### Database
170–223 Raman images per drug and 198–202 RGB images of both sides per drug were taken, respectively. Images for training only, monitoring the training to validate and testing the classification accuracy belong to mutually exclusive group. 100 (118–122), 35–67 (40), and 35–67 (40) for training, validation, and testing CNN for the number of image for the spectral barcode (number of RGB image on each side of drug), respectively. 140-143 images for Raman spectral barcodes of each of the four excluded drugs were used to test the expandability of the spectral barcode.

The drug samples need to be placed at the focal point of the excitation laser. Depending on the position of drug against the objective lens (The aperture size is 0.5 mm in diameter), the sample surface can be either at the focal point or slightly out of focal point. Furthermore, excitation light can be scattered by the etched marks depending on the contacting position of drugs. These can slightly influence the background intensity of the Raman signal. Thus, drugs are placed randomly on the hole of the objective lens to obtain Raman images for training testing CNN.

RGB images were acquired in the normal direction with constraints that the drugs were placed on black paper under typical room light. Preprocessing of the RGB images consisted of denoising, extracting contours, erasing the background, resizing the images, and color normalization. The RGB images for training were augmented to reduce the dependence on the position, angle, and size of the drug in the images.

In this study, the relatively small number of images was sufficient, even though tens of thousands or hundreds of thousands of images are generally used in the artificial intelligence field. That is why there is little chance of other signals in spectral barcodes, and condition of RGB image measurements is limited.

### Integration of smartphone Raman spectrometer
120 band pass filters in the range of 830–910 nm were formed on a quartz substrate by plasma-enhanced chemical vapor deposition and

photolithography. Each band pass consists of one vertical pair of DBR separated with a Si cavity layer, performing as a Fabry–Perot filter. The bottom DBR consists of $TiO_2$ (90 nm)/$SiO_2$ (146 nm)/Si (55 nm)/$SiO_2$ (146 nm)/Si (55 nm). The thickness of each layer of the DBR is determined as $\lambda_{peak}$/4n where $\lambda_{peak}$ is peak wavelength and n is the refractive index of each layer, respectively. For the top DBR, the identical structure of the bottom DBR is inversely stacked in order. The thickness of the Si cavity layer between the DBR layers is determined as $\lambda_{peak}$/2n (thickness range: 300–380 nm) adjusting the peak wavelength, where n is the refractive index of Si. Each filter has a narrow band in the range of 1.0–1.2 nm of full width half maximum (FWHM) and 0.45–0.61 transmission. The difference of the peak-to-peak wavelength ($\Delta\lambda$) was <1.2 nm except two filters for the two longest wavelengths, in which case $\Delta\lambda$ was 2 nm. Distinct 120 filters and 8 Cr metal blocks with image sensor consists 128 CHs, arranged in a 16×8 mosaic pattern. With the image sensor of the rear wide camera in Galaxy Note 9, 4 × 9 array of 128 CHs are manufactured, after removing the camera lens module and the infrared cutoff filter.

### Attachable Raman module

Laser diodes were purchased from Thorlabs, Inc. (LD785-SEV300). Commercially available rechargeable batteries were used as an external power supply. The outer module case works as dark room to block ambient light and pinholes are designed to exclude unwanted scattered light or residual excitation light source. The laser power and the frequency were stable for hours by supplying the external electricity. Also, heat sink was carefully designed in the attachable Raman module to maintain the output wavelength. The laser can be powered by the external power supply such as commercially available rechargeable battery. The rechargeable battery can be as small as to be installed in the attachable Raman module as shown in Fig. 1 or Supplementary Fig. 1. The position of the rear camera and the size of the image sensor can be varied depending on the model of the smartphone. By only modifying the position of outlet of the Raman signal to the image sensor, the optics set-up of the attachable external module can be applicable to other models of the smartphone.

### Progression of image capture with a smartphone Raman spectrometer to the spectral barcode

The size of the image sensor is 3024 pixels × 4032 pixels, and 40 pixels × 40 pixels is for each CH. Among the 4 × 9 array of 128 CHs in the raw image, the data from a 2 × 2 array illuminated with the Raman signal are extracted for further analysis. It is transformed into a spectral barcode after a series of processing, denoising, averaging these 4 sets of 128 CHs, and normalizing. Denoising is performed by averaging values from 20 pixels × 20 pixels of each channel to reduce random noise. The exposure time was set to 10 s in one shot to obtain a weak Raman signal and reduce the readout noise by modifying the AP of a Galaxy Note 9.

The reference Raman spectra from 58 drugs were obtained with a commercial Raman spectrometer equipped with a 785 nm excitation laser (XperRam, Nanobase, Inc.). The spectra were carefully measured to compare intensities in absolute values.

### CNN architecture

To classify the spectral barcode, CNN with a ResNet structure is used. One feature of the ResNet is a shortcut, adding input-to-output after convolution and batch normalization. Because a spectral barcode has only 120 degrees of freedom, only one residual block (convolution a with 3 × 3 kernel size, batch normalization, add, and activation with ReLu) is used. After flattening, two fully connected layers are followed by batch normalization; one with ReLu, and one with softmax as an

activation function. The final output has 11 dimensions, the number of major components.

To identify the brand name of the drug, a classification algorithm in series is designed. There is one CNN for classifying the major component and nine CNNs for identifying the brand name, because there is no need to identify the brand name for vitamins B6 and C. The structure of the CNN for classifying the brand name of the drug is similar to that for the major component. The only difference is the size of the fully connected layer.

The CNN for the RGB images uses the architecture of VGGNet to classify the shape and color. The CNN consists of three convolutional layers with batch normalization, ReLu, and max pooling. After flattening, three fully connected layers are achieved by batch normalization, the last layer with softmax, and the other layers with ReLu as an activation function.

To estimate the accuracy of the combined CNNs for the spectral barcodes and RGB images, a randomly selected spectral barcode and RGB image are tested 1000 times. The result of the accuracy is 83.2% and the deviation of the accuracy is ±0.2%p with 10 trials.

The overfitting is avoided by monitoring the training loss and validation loss simultaneously. As over fitted, validation loss starts to saturate or even increase while training loss keeps decreasing. Therefore, training and validation losses are monitored during training process, as epoch increases. Also overfitting can occur with complicated algorithm structure, and thus the number of hidden layers and parameters needs to be optimized. Furthermore, batch normalization is added after convolution layer and fully-connected layer to prevent gradient vanishing problem which stops updating the parameters in CNN.

### Android application

Android application was produced using program language C#, and installed in galaxy not 9 from Samsung. The drug classification operated by smartphone Raman spectrometer can be confirmed in Supplementary Movie 1 using three drugs (Tylenol, Lipito-M, Diabex) with 5 s of Raman signal collection time.

### Reporting summary

Further information on research design is available in the Nature Portfolio Reporting Summary linked to this article.

## Data availability

Source data and database containing minimum set of spectral barcodes and RGB images of each drugs used in this work have been deposited in the repository[34].

## Code availability

The software with a readme.txt file for installing and running the software can be found in the repository[34].

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

## Acknowledgements

This work was supported National Research Foundation of Korea (NRF-2021R1F1A1062182: Y.P.; NRF- 2020R1A6A1A03047771, Y.P.; NRF-2021R1A2C1010747, H.S.) This work was supported by the Ministry of Health Welfare of Korea (HR21C0885, M.J.C.).

## Author contributions

U.J.K, S.L, M.J.C, H.S., and H.C. conceived and designed the research. U.J.K., S.K and. H.J.K. conducted optical measurements. H.C.K., Y.R., and Y.P. fabricated the filters and measured the corresponding transmissions. S.L. and S.H devised and carried out the CNN. H.S. designed and fabricated the external Raman module. H.J.K. tested the external Raman module. M.J.C. selected the list of drugs and provided the right procedure on treating the drugs during research. U.J.K. and S.L. wrote the manuscript. All authors reviewed the manuscript.

## Competing interests

The authors declare no competing interests.
