## [Peer review file · Nature Communications]

REVIEWER COMMENTS

Reviewer #1 (Remarks to the Author):

In this paper, a novel smartphone raman spectrometer is introduced. Functionality of the measurement principle is demonstrated measuring raman scattering from different drugs while distinguishing different major components and brands. This paper shows a very interesting new approach for measurement of raman scattering, including the use of 120 different filters. The results are very promising. Nevertheless, there are some points that should be revised before publication, including method description, as well as data evaluation and especially discussion.

The CNN has been trained with 54 different drugs and validated with 4 drugs. The described accuracy of your model, which was calculated based on the confusion matrices, was therefore based on investigation of drugs also included in training set. This could result in falsely increased accuracy. Training and validation set should not overlap.

You can measure in the wavelength region from 830 – 910 nm, but in the first chapter, you talk about new peaks at 810, 825 and 830 nm due to additives in drugs which could indicate another brand. Could your precision be improved by addition of these wavelengths? Because smartphone raman spectrometer does not include them. This should be discussed in the paper.

Where are measurement data of the four drugs, not included in the training set? You only have matrices for all drugs which were included but the most important four are missing in all tables. You only talk about accuracy of "99%" and "ca. 100%" but did not discuss anything referring to your validation experiment.

You also say "Increasing the accuracy to ca. 100% is possible by using additional information, such as etch marks or more subdivisions of color or shape, for training the CNN algorithm" which is a very risky statement. You should narrow or relativize this statement because you did not try it and you did not give any reasons to assume that 100% can be achieved.

Your chapter "Discussion" is not a discussion but a conclusion. You do not discuss anything or give new information that has not been written before.

Referring to usability of your device, some important information should be included and discussed:

- Can the setup also be adapted to other smartphones or is the usability restricted to Samsung Galaxy Note 9?
- Did you integrate calculation in a smartphone application or how is it implemented?
- How is the setup powered? Is it powered by the smartphone or is an external power supply needed?
- What about fluctuations in voltage supply- does it influence the measurement results or are there any components to guarantee constant voltage supply?
- There is no information for the used electrical components (for example the laser diode).
- How is "typical roomlight" guaranteed? Is there always the same light source in a dark room or are measurements performed at daylight, for example always at the same time of the day?
- If measurements were performed at daylight- how does changing light composition due to changing time/ weather affect your measurement? Because daylight also includes light in the infrared range which could be reflected by the drugs and measured by the spectrometer. This could affect the baseline which is described to be used for distinguishing between different brands.
- How does distance/ angle/.. between smartphone and drugs influence the Raman measurement? Do you have to be very accurate? In this paper, it is just discussed for RGB images.
- Did you always use the same pill for data acquisition? Does changing of the pill also influence the measurement due to little differences in surface and therefore in light scattering?

Some minor points that should be included:

- In the introduction, the use of heavy numerical calculations for spectra production as a disadvantage for existing Raman spectrometer is described. It is not clear for me, why the production of spectra based on gratings is more complex in comparison to a matrix.
- Be consistent with decimal points. Inconsistency is confusing for the reader. For example, in the abstract, 80 % drug-brand-identification accuracy is described, later it changes to 79.5 %.

Furthermore 99 % accuracy for the drugs that were not included in the training set is changed to ~100% which is not the same.

- In the chapter "Smartphone Raman Spectrometer..." it is described: "The array of 128 filters passes through 128 different wavelengths...", but if I understood it correctly, you have only 120 different filters and wavelengths, because the remaining eight channels are used for metal blocks.

- There should be more discussion about advantages and disadvantages of the system in comparison to other raman spectrometers. What about cost of it? Because the filter production seems to be elaborate. There is already one raman spectrometer available:

<https://labsystematic.com/product/smart-raman-xi/> - are there any advantages? Some disadvantages of existing raman spectrometers are named but there is no section that compares to the developed system.

- The structure of the paper is not completely clear to me. In the first chapter "Smartphone Raman spectrometer and spectral barcode", there is one part "We demonstrated drug classification with a smartphone Raman spectrometer because.." which should be written earlier in the introduction. This does not belong to experimental part. Sometimes, the author switches between different chapters which makes it hard to follow. First, the spectrometer and investigated drugs and data acquisition has been described, including first spectra and fingerprints. Afterwards, there is one chapter about spectral barcodes, but now related to the main components. This chapter is interrupted by one encoding chapter, which is confusing because afterwards, it switches again towards the main component analysis. It would be easier for the reader to change the order to not interrupt chapters and follow one thread. In this case, I would recommend rename of headings.

- "Fig. S4 shows reference Raman spectra measured with a commercial Raman spectrometer at 785-nm excitation." Here I think it should be referred to Fig. S5 instead of Fig. S4.

- Extended Data Fig. 3- y-achsis not consistent.

Reviewer #2 (Remarks to the Author):

Raman spectroscopy is a versatile technique with applications in many disciplines including geology, pharmaceutical manufacturing, medicine, and forensic science, among others. This technique is advantageous as analysis can be performed through packaging, protecting the user from potentially harmful materials, and is generally considered a non-destructive technique. However, it also has limitations, such as fluorescence when analyzing colored materials, and can result in higher false positive rates when analyzing multi component mixtures. This is especially true with handheld/portable Raman instruments. Machine learning and deep learning are becoming prevalent, powerful techniques for drug classification due to their ability to identify spectral differences beyond the capabilities of the human eye. Additionally, several publications have demonstrated that higher accuracies are observed for drug classification when machine learning is implemented in handheld/portable Raman instruments instead of relying on spectral similarity metrics such as cosine similarity, Pearson correlation, and hit-quality-index.

In this article, the authors classified drug with high accuracies by implementing convolutional neural networks in a smartphone Raman spectrometer using a spectral barcode. The approach is relevant in the field as agencies/ companies continue to seek quick, easy, cheap, and accurate methods for drug analysis.

My general comments/ suggestions to the authors include:

- The authors demonstrate novelty by creating their own Raman smartphone for drug analysis. However, the authors should refer to this article: 10.1117/12.2519139 by Chandler, Huang, and Mu to support/ discuss their findings. In the article by Chandler et al, the entire spectrum was used. Here, the authors created a spectral barcode and should consider discussing why this was the selected method.

- Several publications have demonstrated CNNs improve the classification of drugs when analyzed using handheld or portable Raman instruments. Adding a few of these references can help to support the authors findings in this manuscript.

- The authors can also emphasize the uniqueness of this project, in the introduction and conclusion. The reported accuracy is great, but what makes this work different from others? This should be explicitly stated.

- Since the barcoding concept is not novel the authors may consider addressing the advantages of using a barcode vs using the entire spectrum for CNNs? Several other publications tend to use the entire spectrum. I see this as an opportunity for the authors to share the advantage to the reader.

- What measures were taken to avoid overfitting of the CNN models?

- Color and shape for classification of pills may lead to high misidentifications as counterfeiters can easily add dyes/ remove dyes from pills to circumvent detection. This limitation and a solution in future works can be addressed.

- The conclusions are valid. However, I suggest the authors include a few sentences discussing the impact of their work and including some of the limitations of the study.

My comments for the results section is attached.

Suggested improvements:

Comments attached.

- In addition to explaining the need to capture weak Raman signatures and developing a new smartphone Raman spectrometer in the introduction, the authors could have addressed why a smartphone spectrometer integrated with AI capability would be beneficial to various disciplines.

I recommend inviting the authors to revise their manuscript to address key concerns before a final decision is reached.

REVIEWER COMMENTS

Reviewer #1 (Remarks to the Author):

In this paper, a novel smartphone raman spectrometer is introduced. Functionality of the measurement principle is demonstrated measuring raman scattering from different drugs while distinguishing different major components and brands. This paper shows a very interesting new approach for measurement of raman scattering, including the use of 120 different filters. The results are very promising. Nevertheless, there are some points that should be revised before publication, including method description, as well as data evaluation and especially discussion.

Q1)The CNN has been trained with 54 different drugs and validated with 4 drugs. The described accuracy of your model, which was calculated based on the confusion matrices, was therefore based on investigation of drugs also included in training set. This could result in falsely increased accuracy. Training and validation set should not overlap.

Answer) We have to clarify training procedure which may cause misunderstandings. 170~223 Raman images per drug and 198~202 RGB images of both sides per drug were taken, respectively. Images for training only, monitoring the training to validate and testing the classification accuracy belong to mutually exclusive group. To optimize CNN images, one hundred of Raman images and 118 ~ 122 RGB images were used for training, and 35~67 Raman images and 40 RGB images were used for validation while training CNN. And the same number of images of Raman and RGB were used to test the classification accuracy. We added some sentences in Database of Method section to make this issue clearer as shown below.

Database of Method section

“170~223 Raman images per drug and 198~202 RGB images of both sides per drug were taken, respectively. Images for training only, monitoring the training to validate and testing the classification accuracy belong to mutually exclusive group. 100 (118–122), 35–67 (40), and 35–67 (40) for training, validation, and testing CNN for number of image for the spectral barcode (number of RGB image on each side of drug), respectively.”

Q2)You can measure in the wavelength region from 830 – 910 nm, but in the first chapter, you talk about new peaks at 810, 825 and 830 nm due to additives in drugs which could indicate another brand. Could your precision be improved by addition of these wavelengths? Because smartphone raman spectrometer does not include them. This should be discussed in the paper.

Answer) As the referee mentioned, the Raman peaks appeared at 810, 825 and 830nm are associated with additives in the drugs. By increasing the spectral range of smartphone Raman spectrometer to include these wavelengths, the additional spectral information on the drugs could enhance the accuracy to predict band names. We added some discussion in page 8.

Page 8

“It is valuable to mention that broader spectral range could enhance the accuracy to predict its brand names by designing the band pass filter arrays on the image sensor to capture additional Raman features, for example, at 810, 825 and 830 nm associated with additives in the drugs as shown in Supplementary Fig. 5.”

Q3) Where are measurement data of the four drugs, not included in the training set? You only have matrices for all drugs which were included but the most important four are missing in all tables. You only talk about accuracy of “99%” and “ca. 100%” but did not discuss anything referring to your validation experiment.

Answer) Thank you for raising an issue on four drugs which were not included in the training set. Since four drugs are not included in CNN training procedure (both training and validation set), its brand names and outer appearances (color and shape) do not exist in the list or prediction. Thus, our trained CNN cannot provide the answer of its brand name and outer appearance, but its major chemical component. This is the reason that four drugs cannot be included in the confusion matrix, in particular, for brand name. We tested the prediction accuracy for major component for three drugs from metformin and one drug from glimepiride as listed by A1, A2, A3, A4 in Supplementary Table 2 by 424 and 141 trials, respectively. Only one failure of drug from metformin was confirmed, which corresponds to 99.8% accuracy for unknown four drugs. We added some discussion to clarify this issue in page 8.

Page 8

“The prediction accuracy for major component of three drugs from metformin and one drug from glimepiride was obtained from 424 and 141 trials, respectively. Only one failure from metformin was confirmed, which corresponds to 99.8% accuracy for the major component prediction of unknown four drugs.”

Q4) You also say “Increasing the accuracy to ca. 100% is possible by using additional information, such as etch marks or more subdivisions of color or shape, for training the CNN algorithm” which is a very risky statement. You should narrow or relativize this statement because you did not try it and you did not give any reasons to assume that 100% can be achieved.

Answer) We agree with the referee that our statement can be risky without actual trials using etched marks or more subdivisions of color and shape for training the CNN. We modified the original sentence to convey our intention on increasing the accuracy of CNN as shown below.

In Discussion

“There might be methods to increase the prediction accuracy to 100% by using additional information on drugs, such as etched marks or additional spectral features, and even upgrading the specification of the smartphone spectrometers such as spectral range, Q-factor, SNR and etc.”

Q5) Your chapter “Discussion” is not a discussion but a conclusion. You do not discuss anything or give new information that has not been written before.

Answer) We largely modified the discussion section based on the referee’s comments in the discussion section.

Q6) Referring to usability of your device, some important information should be included and discussed: Can the setup also be adapted to other smartphones or is the usability restricted to Samsung Galaxy Note 9?

Answer) The position of the rear camera and the size of the image sensor can be varied depending on the model of the smartphone. By only modifying the position of outlet of the Raman signal to the image sensor, the optics set-up of the attachable external module can be applicable to other models of the smartphone. We added the comments in newly added “Attachable Raman module” section in Methods.

Q7) Did you integrate calculation in a smartphone application or how is it implemented?

Answer) Android application was produced using program language C#, and installed in galaxy note 9 from Samsung. The drug classification operated by smartphone Raman spectrometer can be confirmed as exhibited in Supplementary Movie. We add comments in newly added “Andorid Application” section in Methods.

“Andorid Application” section in Methods

“Android application was produced using program language C#, and installed in galaxy not 9 from Samsung. The drug classification operated by smartphone Raman spectrometer can be confirmed in Supplementary movie using three drugs (Tylenol, Lipito-M, Diabex) with 5 seconds of Raman signal collection time.”

Q8) How is the setup powered? Is it powered by the smartphone or is an external power supply needed?

Answer) The laser can be powered by the external power supply such as commercially available rechargeable battery. The rechargeable battery can be as small as to be installed in the attachable Raman module as shown in Fig. 1 or Supplementary Fig.1. We added the comments in newly added “Attachable Raman module” section in Methods.

Q9) What about fluctuations in voltage supply- does it influence the measurement results or are there any components to guarantee constant voltage supply?

Answer) We have not used any special components to guarantee the fluctuation at a given voltage supply. The laser power and the frequency was stable for hours by supplying the external electricity. There was no noticeable change in the obtained Raman image taken by any kind of external power supply. Furthermore, the spectral barcodes of Raman images were prepared after intensity normalization. Thus, the spectral barcode is independent of the intensity of excitation laser. The output wavelength of laser diode is sensitive to the temperature. Thus, heat sink was carefully designed in the attachable Raman module to maintain the output wavelength. We added the comments in newly added “Attachable Raman module” section in Methods.

Q10) There is no information for the used electrical components (for example the laser diode).

Answer) Laser diodes were purchased from Thorlabs, Inc (LD785-SEV300). Commercially available rechargeable batteries were used as external power supply. We added the comments in newly added “Attachable Raman module” section in Methods.

Q11) How is “typical roomlight” guaranteed? Is there always the same light source in a dark room or are measurements performed at daylight, for example always at the same time of the day?

If measurements were performed at daylight- how does changing light composition due to changing time/ weather affect your measurement? Because daylight also includes light in the infrared range which could be reflected by the drugs and measured by the spectrometer. This could affect the baseline which is described to be used for distinguishing between different brands.

Answer) Raman signal is very weak compared to ambient light and excitation light source of which complete rejection is the first step to measure Raman signal successfully. As shown in Fig. 1, outer module case works as dark room to block ambient light. Also, pinholes are specially designed inside the module to exclude unwanted scattered light or residual excitation light source. With ambient light and noise robust attachable module, the Raman signal measurement was conducted reliably and reproducibly regardless of weather, time and location. We are strongly convinced that the baseline of each drugs are associated not with infiltrated ambient light but with its intrinsic optical properties. We added the comments in newly added “Attachable Raman module” section in Methods.

Q12) How does distance/ angle/.. between smartphone and drugs influence the Raman measurement? Do you have to be very accurate? In this paper, it is just discussed for RGB images.

Answer) We appreciate the referee’s comment. The drug samples need to be placed at the focal point of the excitation laser. Our set up is designed to have the focal point at the very end of the objective lens. Thus, Raman signal was collected by contacting the sample at the objective lens, which is not explained in the manuscript clearly.

The measurement and analysis procedure can be confirmed in the Supplementary Movie. But drugs have etched marks, and flat or curved surface. Depending on the position of drug against the objective lens (The aperture size is 0.5 mm in diameter), the sample surface can be either at the focal point or slightly out of focal point. Furthermore, excitation light can be scattered by the etched marks depending on the contacting position of drugs at the objective lens. These can slightly influence the background intensity of the Raman signal. Thus, we obtained Raman images for training and testing CNN by placing the drugs randomly on the hole of objective lens.

The statistical analysis of 42 Raman spectral images of Vitamin C to test CNN has been done as shown below. The average value with standard deviation of normalized Raman intensity at each wavelength of the spectral barcodes is plotted.

We added some discussions on this issue in Method and page 6, and a separate figure as the Supplementary Fig. 6.

Database in Method section

“The drug samples need to be placed at the focal point of the excitation laser. Depending on the position of drug against the objective lens (The aperture size is 0.5 mm in diameter), the sample surface can be either at the focal point or slightly out of focal point. Furthermore, excitation light can be scattered by the etched marks depending on the contacting position of drugs. These can slightly influence the background intensity of the Raman signal. Thus, drugs are placed randomly on the hole of objective lens to obtain Raman images for training testing CNN.”

Page 6

“The details to obtain Raman spectral images are explained in the Methods. The statistical analysis of 42 Raman spectral images of Vitamin C to test CNN has been done as shown in Supplementary Fig. 6. The average value with standard deviation of normalized Raman intensity at each wavelength of the spectral barcodes is plotted.”

Q13) Did you always use the same pill for data acquisition? Does changing of the pill also influence the measurement due to little differences in surface and therefore in light scattering?

Answer) We have used a number of pills for data acquisition. We have not noticed any difference originated from little differences in surface but the focusing position dependence within the same pill was observed because of etched marks and curved surface as explained in Q12.

Some minor points that should be included:

- In the introduction, the use of heavy numerical calculations for spectra production as a disadvantage for existing Raman spectrometer is described. It is not clear for me, why the production of spectra based on gratings is more complex in comparison to a matrix.

Answer) We apologize the misleading sentences. Gratings is an excellent optical component in spectrometer to disperse optical signals with high spectral resolution, but is not easy to minimize its form factor to fit into smartphone. To overcome this issue, mini spectrometers by replacing conventional grating by such as photonic crystals, metasurfaces, quantum dots and silicone nanowires integrated on CCD or CMOS image sensors have been investigated. To calculate the input spectrum, $s(\lambda)$ out of measured intensity, $I(\mathbf{x})$ at the detector, numerical analysis may need to be done as expressed by the equation below due to its low Q-factor or complicated form of response function, $r(\lambda, \mathbf{x})$ at each channel where \mathbf{x} is the position of each channel at the detector.

$$I(\mathbf{x}) = \int r(\lambda, \mathbf{x}) \cdot s(\lambda) d\lambda$$

We modified some sentences in introduction to avoid misunderstanding on this issue.

Introduction

“Gratings is an excellent optical component in spectrometer to disperse optical signals with high spectral resolution, but is not easy to minimize its form factor to fit into smartphone. To overcome this issue, mini spectrometers by replacing conventional grating with such as photonic crystals^{14,15}, metasurfaces¹⁶⁻¹⁸, quantum dots¹⁹ and silicone nanowires²⁰ integrated on charge coupled detector (CCD) or CMOS image sensors have been investigated. To calculate the input spectrum, $s(\lambda)$ out of measured intensity, $I(\mathbf{x})$ at the detector, numerical analysis needs to be done as expressed by the equation below due to its low Q-factor or complicated form of response function, $r(\lambda, \mathbf{x})$ at each pixel where \mathbf{x} is the position of each pixel at the detector.

$$I(\mathbf{x}) = \int r(\lambda, \mathbf{x}) \cdot s(\lambda) d\lambda$$

Thus, experimental results in the literature¹⁴⁻²⁰ have substantial limitations—especially in terms of capturing weak and high spectral resolution required for Raman signatures.”

- Be consistent with decimal points. Inconsistency is confusing for the reader. For example, in the abstract, 80 % drug-brand-identification accuracy is described, later it changes to 79.5 %. Furthermore 99 % accuracy for the drugs that were not included in the training set is changed to ~100% which is not the same.

Answer) We fixed the number for prediction accuracy to be consistent over the entire manuscript. The prediction accuracy for major component and brand name by only spectral barcode is 99.0% and 79.5%, respectively. The prediction accuracy for color and shape by CNN of RGB images is now 95.6%, and that for brand name by combined CNN of RGB and spectral barcode is 83.2%. The prediction accuracy for major component for four drugs which is not included in the training set, is 99.8% (only one failure out of 424 trials).

- In the chapter "Smartphone Raman Spectrometer..." it is described: "The array of 128 filters passes through 128 different wavelengths...", but if I understood it correctly, you have only 120 different filters and wavelengths, because the remaining eight channels are used for metal blocks.
-

Answer) As the referee mentioned, strictly speaking, the number of filter is 120. In 120 channels out of 128 channels, filters pass through 120 distinct wavelengths in the range of 830~910 nm. Remaining 8 channels are blocked by Cr layer used as position indicator. We modified and added some sentences to clear this uncertainty in page 4.

Page 4

“For 120 CHs out of 128 CHs, its band pass filters transmit 120 distinct wavelengths in the range of 830~910 nm. The rest CHs are blocked by metal as position indicators exhibited as black squares in Fig. 1.”

- There should be more discussion about advantages and disadvantages of the system in comparison to other raman spectrometers. What about cost of it? Because the filter production seems to be elaborate. There is already one raman spectrometer available: <https://labsystematic.com/product/smart-raman-xi/> are there any advantages? Some disadvantages of existing raman spectrometers are named but there is no section that compares to the developed system.

Answer) Thank you very much for the comment. We agree with the referee that comparison with the existing Raman spectrometers with our current system must be performed. One of the existing smartphone based Raman spectrometers, CloudMinds's XITM, is working by attaching external Raman spectrometer including the laser diode through the electronic control board to the smartphone. After taking Raman spectrum by an android smartphone application, the smartphone connects XITM to the cloud to analyze the data by AI deep learning algorithm. In this work, the image sensor of the smartphone camera has been developed into spectrometers by forming Fabry-Perot filter arrays. Only external excitation and collection optics are needed to excite and collect Raman signals from the specimen under interest without additional connecting electronic board to the smartphone. Thus, the form factor of the attachable Raman module is much smaller than XITM. The acquired Raman signal is analyzed by CNN working on-device without connecting it to external cloud system. Since XITM is integrated with conventional grating, its spectral resolution (<1 nm) is better than that of the current work (1~1.2 nm by FWHM). XITM provides high precision spectral information on the samples, but still that of developed spectrometer in this work is high enough to measure Raman spectrum. The developed spectrometer produces lower data size which enables to consume less computing resources favorable to data management in smartphone. The spectral range of this work can be somewhat lower than that of previous works at this point, and this can be overcome by designing the wavelength set of filters and/or increasing number of channels in the array. There is a drawback of the developed spectrometer that changing the excitation wavelength for Raman measurement is limited due to the fixed set of filter wavelengths. Thus, allowed wavelengths of excitation laser should be shorter than maximum wavelength of filters, respectively. The production cost of filter arrays on the image sensor must be very high due to its elaborate fabrication process. Among the existing and developed Raman spectrometers, it is not easy to comment on the cost at this point because the price of image sensor based spectrometer can be extensively lowered by mass production. We summarized few works demonstrating smartphone based spectrometers in the literatures as a separated table (Supplementary Table 1) and added detailed comparison as its table caption, and added some discussions in page 4.

	Spectral range	Spectral resolution	Form Factor	Optical Source	Detector	Spectrometer	Data analysis
CloudMinds's XI™	798~914 nm	< 1 nm	159×79×27 mm (without case)	785 nm LD	CMOS	External spectrometer	CNN based algorithm working at cloud
Changhong's H2	near IR	-	-	-	CMOS	Imbedded spectrometer of Scio by Consumer Physics, Inc.	-
GoyaLab's IndiGo UV/VIS	380~720 nm	<1.5 nm (FWHM)	76×45×53 mm	-	CMOS	External spectrometer	-
Current work	830~910 nm	1~1.2 nm (FWHM)	83×42×18 mm	785 nm LD	CMOS	Image sensor based spectrometer	CNN based algorithm installed in On-device

Supplementary Table 1. Comparison of smartphone based spectrometers. All of the works on smartphone based spectrometers demonstrate either the image sensor of the smartphone camera as detector, or smartphone as electronic controller and/or communication platform of external/installed spectrometer modules. One of the existing smartphone based Raman spectrometers, CloudMinds's XI™, is working by attaching external Raman spectrometer including the laser diode (LD) through the extra electronic control board to the smartphone. After taking Raman spectrum, the smartphone connects the cloud to analyze the data by deep learning algorithm. Changhong's H2 with a miniaturized and integrated material sensor was introduced at consumer electronics show in 2017. Material sensing near IR spectrometer (SCiO from Consumer physics, Inc.) was integrated in the smartphone (not image sensor of smartphone camera) in collaboration with Analog Devices Inc. Also, stand-alone miniaturized spectrometers with high spectral resolution which is controllable by android smartphones are commercially available, for example, Indigo UV/VIS from GoyaLab. In this work, the image sensor of the smartphone camera has been developed into spectrometers by forming Fabry-Perot filter arrays. Only external excitation and collection optics are needed to excite and collect Raman signals without additional connecting electronic board to the smartphone. This makes the smartphone spectrometer more compact with minimized external module and versatile. Higher spectral resolution can be acquired using grating type spectrometer, such as XI™, but still that of developed spectrometer in this work is high enough to measure Raman spectrum, producing lower data size which is favorable to data management in smartphone. Acquiring the spectrum of the objects can be done by the smartphone application, and further analysis can be done using CNN by on-device or connecting to the cloud. The spectral range of this work can be somewhat lower than that of previous works at this point, and this can be overcome by designing the wavelength set of filters and/or increasing number of channels in the array. There is a drawback of the developed spectrometer that changing the excitation wavelength for Raman measurement is limited due to the fixed set of filter wavelengths since the allowed wavelengths of excitation laser should be shorter than maximum wavelength of filters, respectively.

- The structure of the paper is not completely clear to me. In the first chapter "Smartphone Raman spectrometer and spectral barcode", there is one part "We demonstrated drug classification with a smartphone Raman spectrometer because.." which should be written earlier in the introduction. This does not belong to experimental part. Sometimes, the author switches between different chapters which makes it hard to follow. First, the spectrometer and investigated drugs and data acquisition has been described, including first spectra and fingerprints. Afterwards, there is one chapter

about spectral barcodes, but now related to the main components. This chapter is interrupted by one encoding chapter, which is confusing because afterwards, it switches again towards the main component analysis. It would be easier for the reader to change the order to not interrupt chapters and follow one thread. In this case, I would recommend rename of headings.

Answer) We tried hard to modify the structure of the manuscript extensively by moving some sentences in the main part to the introduction and by renaming headings.

- “Fig. S4 shows reference Raman spectra measured with a commercial Raman spectrometer at 785-nm excitation.”

Here I think it should be referred to Fig. S5 instead of Fig. S4.

Answer) We now replaced Fig.S4 by “Supplementary Fig.5”.

- Extended Data Fig. 3- y-axis not consistent.

Answer) We now corrected the y-axis of Supplementary Fig. 3 to be consistent.

Reviewer #2 (Remarks to the Author):

Raman spectroscopy is a versatile technique with applications in many disciplines including geology, pharmaceutical manufacturing, medicine, and forensic science, among others. This technique is advantageous as analysis can be performed through packaging, protecting the user from potentially harmful materials, and is generally considered a non-destructive technique. However, it also has limitations, such as fluorescence when analyzing colored materials, and can result in higher false positive rates when analyzing multi component mixtures. This is especially true with handheld/ portable Raman instruments. Machine learning and deep learning are becoming prevalent, powerful techniques for drug classification due to their ability to identify spectral differences beyond the capabilities of the human eye. Additionally, several publications have demonstrated that higher accuracies are observed for drug classification when machine learning is implemented in handheld/ portable Raman instruments instead of relying on spectral similarity metrics such as cosine similarity, Pearson correlation, and hit-quality-index.

In this article, the authors classified drug with high accuracies by implementing convolutional neural networks in a smartphone Raman spectrometer using a spectral barcode. The approach is relevant in the field as agencies/ companies continue to seek quick, easy, cheap, and accurate methods for drug analysis.

My general comments/ suggestions to the authors include:

- The authors demonstrate novelty by creating their own Raman smartphone for drug analysis. However, the authors should refer to this article: 10.1117/12.2519139 by Chandler, Huang, and Mu to support/ discuss their findings. In the article by Chandler et al, the entire spectrum was used. Here, the authors created a spectral barcode and should consider discussing why this was the selected method.

Answer) Thank you very much for the comment. We agree with the referee that comparison with the existing Raman spectrometers with our current system must be performed. One of the existing smartphone based Raman spectrometers, CloudMinds's XITM, is working by attaching external Raman spectrometer including the laser diode through the electronic control board to the smartphone. After taking Raman spectrum by an android smartphone application, the smartphone connects XITM to the cloud to analyze the data by AI deep learning algorithm. In this work, the image sensor of the smartphone camera has been developed into spectrometers by forming Fabry-Perot filter arrays. Only external excitation and collection optics are needed to excite and collect Raman signals from the specimen under interest without additional connecting electronic board to the smartphone. Thus, the form factor of the attachable Raman module is much smaller than XITM. The acquired Raman signal is analyzed by CNN working on-device without connecting it to external cloud system. Since XITM is integrated with conventional grating, its spectral resolution (<1 nm) is better than that of the current work (1~1.2 nm by FWHM). XITM provides high precision spectral information on the samples, but still that of developed spectrometer in this work is high enough to measure Raman spectrum, *i. e.* spectral barcode (see Supplementary Table 5). The developed spectrometer produces lower data size which enables to consume less computing resources favorable to data management in smartphone. We summarized few works to demonstrating smartphone based spectrometers in the literatures as a separated table (Supplementary Table 1), and added some discussions in page 4.

“All of the works on smartphone based spectrometers demonstrate either the image sensor of the smartphone camera as detector, or smartphone as electronic controller and/or communication platform of external/installed spectrometer modules. One of the existing smartphone based Raman spectrometers, CloudMinds's XITM, is working by attaching external Raman spectrometer including the laser diode (LD) through the extra electronic control board to the smartphone. After taking Raman spectrum, the smartphone connects the cloud to analyze the data by deep learning algorithm. Changhong's H2 with a miniaturized and integrated material sensor was introduced at consumer electronics show in 2017. Material sensing near IR spectrometer (SCiO from Consumer physics, Inc.) was integrated in the smartphone (not image sensor of smartphone camera) in collaboration with Analog Devices Inc. Also, stand-alone miniaturized spectrometers with high spectral resolution which is controllable by android smartphones are commercially available, for example, Indigo UV/VIS from GoyaLab. In this work, the image sensor of the smartphone

camera has been developed into spectrometers by forming Fabry-Perot filter arrays. Only external excitation and collection optics are needed to excite and collect Raman signals without additional connecting electronic board to the smartphone. This makes the smartphone spectrometer more compact with minimized external module and versatile. Higher spectral resolution can be acquired using grating type spectrometer, such as XITM, but still that of developed spectrometer in this work is high enough to measure Raman spectrum, producing lower data size which is favorable to data management in smartphone. Acquiring the spectrum of the objects can be done by the smartphone application, and further analysis can be done using CNN by on-device or connecting to the cloud.”

	Spectral range	Spectral resolution	Form Factor	Optical Source	Detector	Spectrometer	Data analysis	
CloudMinds's XITM	798~914 nm	< 1 nm	159×79×27 mm (without case)	785 nm LD	CMOS	External spectrometer	CNN based algorithm working at cloud	
Changhong's H2	near IR	-	-	-	CMOS	Imbedded spectrometer of Scio Consumer Physics, Inc. by	-	
GoyaLab's IndiGo UV/VIS	380~720 nm	<1.5 nm (FWHM)	76×45×53 mm	-	CMOS	External spectrometer	-	
Current work	830~910 nm	1~1.2 nm (FWHM)	83×42×18 mm	785 nm LD	CMOS	Image sensor based spectrometer	CNN based algorithm installed in On-device	

- Several publications have demonstrated CNNs improve the classification of drugs when analyzed using handheld or portable Raman instruments. Adding a few of these references can help to support the authors findings in this manuscript.

Answer) We agree with the referee that by adding some references on classification of drugs using Raman spectrometer with machine learning, the current work can be supported. There have been some researches in the literature on classifying drugs by Raman spectroscopy with machine learning²²⁻²⁶. Classifying pharmaceutical ingredients, and detection of newly emerging psychoactive substance and illicit drugs were demonstrated by partial least squares-discriminate analysis (PLS-DA)²², principal component analysis (PCA)²³ and CNN²⁴, respectively. Detection of illicit drugs²⁵ or psychoactive drugs²⁶ were demonstrated even in human urine and finger marks to prevent patients from overdose or misuse of it by support vector machines (SVM) and PLS-DA, respectively.

22. Maltaş, D.C., Kwok, K., Wang, P., Taylor, L.S., Ben-Amotz, D. Rapid classification of pharmaceutical ingredients with Raman spectroscopy using compressive detection strategy with PLS-DA multivariate filters. *J. Pharm. and Biome. Anal.* 20, 63-68 (2013).

23. Calvo-Castro, J. Guirguis, A., Samaras, E. G., Zloh, M., Kirton, S. B. & Stair, J.L. Detection of newly emerging psychoactive substances using Raman spectroscopy and chemometrics. *RSC Adv.* 8, 31924-31933 (2018).

24. Lai, Y-T., Wei, P-K., Kuo, C-Y. & Chen, J-Y. Inference detection and classification of illicit drugs by a modest Raman spectrometer with a convolutional neural network analyzer. *Sen. and Actu. B. Chem.*, 375, 132923 (2023).
25. Dong, R. Weng, S. Yang, L. & Liu, Detection and Direct Readout of Drugs in Human Urine Using Dynamic Surface-Enhanced Raman Spectroscopy and Support Vector Machines. *J. Anal. Chem.* 87, 2937–2944 (2015).
26. Amin, M. O., Al-Hetlani, E. & Lednev, I. K. Detection and identification of drug traces in latent fingerprints using Raman spectroscopy, *Sci. Rep.* 12, 3136 (2022) We added following sentences in the introduction with new references.

Introduction,

“there have been some researches in the literature on classifying drugs by Raman spectroscopy with the aid of machine learning²²⁻²⁶. Classifying pharmaceutical ingredients, and detection of newly emerging psychoactive substance and illicit drugs were demonstrated by partial least squares-discriminate analysis (PLS-DA)²², principal component analysis (PCA)²³ and CNN²⁴, respectively. Detection of illicit drugs²⁵ or psychoactive drugs²⁶ were demonstrated even in human urine and finger marks to prevent patents from overdose or misuse of it by support vector machines (SVM) and PLS-DA, respectively.”

- The authors can also emphasize the uniqueness of this project, in the introduction and conclusion. The reported accuracy is great, but what makes this work different from others? This should be explicitly stated.

Answer) There has been great effort to utilize smartphone in the spectroscopic system by using its image sensor of camera as detector or using it as controlling and analyzing platform of the portable spectrometer and its spectrum. However, in our work, the image sensor of the camera was developed to function as spectrometer without grating in the smartphone. This makes the smartphone spectrometer more compact with minimized external module and versatile. The spectral resolution is quiet lower than conventional dispersive spectrometer with grating, but is enough to classify drugs by Raman spectral barcode by CNN.

We added modified the sentences in discussion section to emphasize the uniqueness and importance of this work.

“In this work, we introduced the concept of the spectral barcode, obtained with a smartphone Raman spectrometer. Even with relatively lower spectral resolution and SNR due to the inherent properties of band pass filter arrays and CMOS image sensor compared with commercially available spectrometers installed with grating and CCD, the smartphone Raman spectrometer exhibits still high enough Q factor as portable spectrometer with high efficiency in terms of power consumption. Only external excitation and collection optics are needed to excite and collect Raman signals from the specimen without additional connecting electronic board to the smartphone. This makes the smartphone spectrometer more compact with minimized external module and versatile. Integrating with AI capability in the smartphone spectrometer makes the developed spectrometer more powerful.”

- Since the barcoding concept is not novel the authors may consider addressing the advantages of using a barcode vs using the entire spectrum for CNNs? Several other publications tend to use the entire spectrum. I see this as an opportunity for the authors to share the advantage to the reader.

Answer) As referee suggested, addressing the advantages of using a barcode and using the entire spectrum for CNN provides the good opportunity to emphasize the advantages of our work. Classifications on various applications by CNN have been done using full spectrum of objects under interest obtained by benchtop or portable spectrometers using high signal to noise ratio (SNR) charge coupled detector (CCD) with high spectral resolution. The smartphone based Raman spectrometer using 128CH filter arrays produces lower spectral resolution but still high Q factor (> 1nm by FWHM). Since in the current work, one channel is consisted of 10×10 pixels, SNR can be further increased by a numerically denoising process. Thus, the developed spectrometer on CMOS image sensor exhibits SNR and Q

factor enough to classify drugs by Raman spectral barcode and lowered power consumption. We added above discussion.

Page 8

“Classifications on various applications by CNN have been done using full spectrum of objects under interest obtained by benchtop or portable spectrometers using high signal to noise ratio (SNR) CCD and conventional grating^{22,26} with high spectral resolution. The smartphone based Raman spectrometer using 120 filter arrays on CMOS image sensor produces lower spectral resolution but still high Q factor (> 1 nm by FWHM). CMOS image sensors are highly efficient in power consumptions compared to CCD. Thus, the developed spectrometer on CMOS image sensor exhibits SNR and Q factor enough to classify drugs by Raman spectral barcode, and is suitable for lower power consumption.”

- What measures were taken to avoid overfitting of the CNN models?

To train CNN, obtaining the best accuracy is the key point while avoiding the overfitting problem. First, the obtained database was saved and managed for training, validation and test mutually exclusively. Since overfitting can occur with complicated algorithm structure, and thus the number of hidden layers and parameters needs to be optimized. The overfitting was avoided by monitoring the training loss and validation loss simultaneously on increasing the number of epoch. When over fitted, validation loss starts to saturate or even increase while training loss keeps decreasing. Furthermore, batch normalization was added after convolution layer and fully-connected layer to prevent gradient vanishing problem which stops updating the parameters in CNN.

We added some sentences in the CNN architecture of the Method section.

CNN architecture in Methods

“The overfitting is avoided by monitoring the training loss and validation loss simultaneously. As over fitted, validation loss starts to saturate or even increase while training loss keeps decreasing. Therefore, training and validation losses are monitored during training process, as epoch increases. Also, overfitting can occur with complicated algorithm structure, and thus the number of hidden layers and parameters needs to be optimized. Furthermore, batch normalization is added after convolution layer and fully-connected layer to prevent gradient vanishing problem which stops updating the parameters in CNN.”

- Color and shape for classification of pills may lead to high misidentifications as counterfeiters can easily add dyes/ remove dyes from pills to circumvent detection. This limitation and a solution in future works can be addressed.

Answer) We agree with the referee that identifying drugs only using its color and shape is very dangerous. This is why the Raman spectrum becomes important for drug classification which is finger print of molecules. In addition, color and shape was treated as supplementary information on drugs. To make the best use of the Raman spectrum on drug classification, extensive works need to be done in the future. In reality, the intensity of Raman spectrum of the major components could be very weak, or the thickness of coating material is too thick that the internal major components cannot be detected. Furthermore, drugs can be produced by mixing several major components, and its analysis can become more complicated with high fluorescence component included. To generalize the current method to the real drug classification, signal to noise ratio of smartphone Raman spectrometer needs to be improved, and detection of major components under thick coating, for example by introducing spatially offset Raman spectroscopy (SORS), may be possible, and fluorescence could be separated from Raman signals by shifted-excitation Raman difference spectroscopy (SERDS). Moreover, systematic understanding how drug companies mix major components in collaboration with medical society is necessary to develop powerful drug classification CNN. We added some sentences in Discussion.

Discussion

“In the measurement aspect for prediction accuracy enhancement, detection of major components under thick coating may be possible, for example, by introducing spatially offset Raman spectroscopy (SORS). Fluorescence could be separated from Raman signals of drugs by shifted-excitation Raman difference spectroscopy (SERDS). Moreover, systematic understanding how drug companies mix major components in collaboration with medical society is necessary to develop more powerful drug classification CNN.”

- The conclusions are valid. However, I suggest the authors include a few sentences discussing the impact of their work and including some of the limitations of the study.

Answer) In this work, we demonstrated that 2D Raman images can be captured by smartphone Raman spectrometer. By reducing the size of channel to one-pixel level and increasing the density of channel arrays, spectral and morphological information of the object under interest at the same time, which is called hyperspectral imaging, by using smartphone camera. This will increase the portability and usability opening up new field smartphone business. We modified and added some sentences at the end of Discussion.

Discussion

“In the future, by reducing the size of channel to one-pixel level and increasing the density of CH arrays, simultaneous measurement of spectral and morphological information of the object under interest can be achieved, which is called hyperspectral imaging, by using smartphone camera. This will extensively increase the portability and usability opening up new field in smartphone business.”

My comments for the results section is attached.

Suggested improvements:

Comments attached.

- In addition to explaining the need to capture weak Raman signatures and developing a new smartphone Raman spectrometer in the introduction, the authors could have addressed why a smartphone spectrometer integrated with AI capability would be beneficial to various disciplines.

Answer) Thank you very much for the comments to provide us to considering the impact of our work in many aspects. Developing a new smartphone Raman spectrometer to capture weak Raman spectrum is one of the main subjects in this work. In addition to that, integrating with AI capability in the smartphone spectrometer is another important subject. Ability to analyze the spectrum taken by the spectrometer enhance its the portability and usability in various disciplines including drug classification in this work. We added some sentences to emphasize the importance of AI integrated smartphone Raman spectrometer at the end of introduction.

Introduction

“Integrating with AI capability in the smartphone spectrometer allows users to analyze the spectrum at various places and situations. This will enhance its portability and usability of smartphone spectrometer in numerous disciplines including drug classification.”

I recommend inviting the authors to revise their manuscript to address key concerns before a final decision is reached.

REVIEWERS' COMMENTS

Reviewer #1 (Remarks to the Author):

The authors have revised the manuscript very carefully. They have sufficiently addressed all the reviewers' comments and have answered the open questions conclusively. I therefore recommend acceptance of the manuscript and have no further requests for changes.

Reviewer #2 (Remarks to the Author):

The authors have done a wonderful job addressing all concerns from the initial manuscript submission.